# Changes in lung function and dyspnea perception in Colombian Covid-19 patients after a 12-week pulmonary rehabilitation program

Carlos D. Páez-Mora[1], Diana Carolina Zona[1]*, Teddy Angarita-Sierra[2], Matilde E. Rojas-Paredes[1], Daniela Cano-Trejos[1]

1 Cuidado Cardiorrespiratorio Research Group, Universidad Manuela Beltrán, Bogotá, Colombia, 2 Grupo de investigación Biodiversidad para la Sociedad, Dirección Académica, Universidad Nacional de Colombia, Sede De La Paz, La Paz, Cesar, Colombia

* dczonar@unal.edu.co

## Abstract

### Background

Although moderate and severe Covid-19 patients have shown obstructive and restrictive disorders in pulmonary function after recovery from the disease, studies evaluating the effectiveness of rehabilitation programs that seek to improve lung function are scarce.

### Aim

Herein, we evaluate changes in lung function and perceived dyspnea in Covid-19 patients after undergoing 12 weeks of a pulmonary rehabilitation (PR) program.

### Design

Retrospective observational study.

### Setting

Cesar, Colombia Neumocesar Pneumological Center.

### Population

100 outpatients with a history of Covid-19.

### Methods

Respiratory function using spirometry parameters, as well as perceived dyspnea, measured by the modified Medical Research Council (mMRC) dyspnea scale, was evaluated in 100 patients with a history of Covid-19. We used univariate and multivariate statistical approaches to assess changes in lung function and perceived dyspnea before and after a PR program to determine whether gender, age, height, weight, comorbidities, and oxygen delivery system affects the recovery of lung function and perceived dyspnea.

**Funding:** The author(s) received no specific funding for this work.

**Competing interests:** The authors have declared that no competing interests exist.

## Results

It was found that PR treatment has positive effects on respiratory pathologies caused by SARS-CoV-2 infection regardless of patient gender (S = 0,029), indicating that rehabilitation provided benefits regardless of the physical characteristics of the patients. Both univariate and multivariate statistical analyses indicated that FVC (P = 0,0001), FEV1(P = 0,0001), and mMRC (P = 0,0001) are robust diagnostic indicators of lung function recovery and perceived dyspnea. Both invasive and non-invasive positive pressure ventilatory support had deleterious effects on lung function prolongating patient recovery (P = 0,0001).

## Conclusions

Rehabilitation programs can benefit patients facing respiratory pathologies caused by SARS-CoV-2 infection. Additional research on the long-term effects of the sequelae of Covid-19 is needed.

## Clinical rehabilitation impact

PR programs have positive effects on patients facing respiratory pathologies caused by SARS-CoV-2 infection.

## Introduction

Coronavirus disease (Covid-19) is an infectious disease caused by the SARS-CoV-2 virus. Many infected people develop mild to moderate respiratory illness and recover without requiring special treatment. However, approximately 20% of Covid-19 patients require medical attention [1]. Peramo et al. [2] demonstrated that the persistence of some symptoms after recovery from the disease are sequelae associated with the infection. The destruction of the alveolar epithelium, lung consolidation, and extensive injury to alveolar epithelial cells are some of the pathophysiological manifestations observed [2]. The deterioration of lung function has been observed after hospitalization due to the evolution of lung injury [3].

Disease progression leads to lung tissue damage, which stems from extensive injury to alveolar epithelial cells following infection. These alterations are more frequent in patients under mechanical ventilation. However, the presence of pulmonary alterations has been documented in pulmonary function tests of apparently healthy patients diagnosed with Covid-19 [2, 4].

Spirometry is the best-known and low-cost respiratory function test [5]. The most commonly used parameters in this test are forced vital capacity (FVC), forced expiratory volume in the first second (FEV1), and the FEV1/FVC ratio [6].

Studies evaluating the lung function of patients after Covid-19 infection are scarce. However, Torres Castro et al. [3] reported obstructive and restrictive disorders of the lung in both moderate and severe Covid-19 patients using spirometry. These authors also evaluated diffusion disorders using the diffusing capacity for carbon monoxide test (DLCO) and found that these were the most frequent alterations observed, even in patients with mild cases and patients without symptoms [7].

Approximately 40% of hospitalized Covid-19 patients have impaired lung function [8]. This lesion has even been observed in patients with mild disease. Restrictive patterns are frequently observed between weeks 6 and 8 after discharge according to pulmonary function tests [9, 10].

PR is a personalized multidisciplinary treatment that seeks to improve the physical and psychological conditions of patients with respiratory diseases. It has been used worldwide in patients with lung diseases such as chronic obstructive pulmonary disease (COPD) and can significantly improve lung function and quality of life. [10, 11]. It has also been used on patients in the recovery phase of Covid-19 [8–10].

Evaluations of lung function and dyspnea perception in patients undergoing PR are needed to improve our understanding of the sequelae of Covid-19 [12]. Due to the scarcity of evidence-based guidelines regarding rehabilitation after Covid-19, there is an urgent need for studies to evaluate the lung function and dyspnea perception before and after PR programs [10, 13, 14]. The Stanford Hall consensus for post-Covid-19 rehabilitation highlighted the need for more research on the sequelae of the disease and its long-term impact [10].

The aim of this study was to assess changes in lung function and perceived dyspnea in a group of patients diagnosed with Covid-19 after undergoing a 12-week PR program in a clinic in Colombia.

## Materials and methods

### Study design

Retrospective observational study was carried out and pre- and post-intervention lung function was evaluated.

### Setting and patients

This study was conducted at NeumoCesar Pulmonary Center, Valledupar, Colombia. Approval was obtained from the ethics committee of Manuela Beltrán University in Bogotá, Colombia (TRCC-CE-I-2210009). The sample size was estimated using Slovin's formula as follows:

$$n = \frac{N}{1 + Ne^2}$$

where n is the sample size, $N$ is the total number of Covid-19 patients attended at NeumoCesar who received therapy (231 patients in this case), $e$ is the maximum variability or margin of error (7.5% or 0.07), and 1 is the probability of an event occurring. Resolving Slovin's formula, the sample size obtained was 100. Therefore, 100 patients diagnosed with Covid-19, who completed the PR program in 2022 and met the following inclusion and exclusion criteria, were included:

Inclusion criteria:

- Post-covid-19 patients

- Patients had been discharged from the hospital

- Patients with 100% compliance with the PR program at the NeumoCesar institution.

  Exclusion criteria:

- Serious arrhythmias, syncope induced by exercise or decompensated metabolic disorders without adequate management

- Coexistence of musculoskeletal or neurological problems that reduce mobility

- History of severe psychiatric disorder, cognitive disorders, or organic mental syndrome that prevents interaction or understanding instructions.

## Intervention

**Pre.**   Spirometry was performed, sociodemographic data were recorded, and FVC, FEV1, FEV1/FVC, forced expiratory flow (FEF) 25–75, and peak expiratory flow (PEF) were evaluated; dyspnea was evaluated using the mMRC scale.

The PR program followed the recommendations of a previous publication of the Colombian Ministry of Health and Social Protection entitled "Interventions for a Pulmonary Rehabilitation Program" [15]. This is a conventional rehabilitation program, however, was tailored to each patient's individual needs and tolerance. During the sessions the modified BORG scale was used to assess the patient´s effort. Intensity was gradually adjusted by a therapist over the course of the rehabilitation in each session. Intensity was defined according to hypoxemia, severity of the disease, age, among other variables. Patients performed cardiorespiratory endurance training, accomplishing high-intensity intervals of 2 to 3 minutes with 60 to 80% of the maximum cardiac capacity, the intensity of the exercise was severe. The oxygen saturation of the patients was monitored with a pulse oximeter ensuring that the saturation range was always above 90%.

The rehabilitation program included different components such as pharmacological management, patient and family education, nutritional follow-up, psychosocial support, daily living aid training, vocational counseling, sexuality, and energy conservation techniques. These components were led by a multidisciplinary team that included a Doctor, Respiratory Therapist, Physiotherapist, Nurse, Nutritionist, Occupational Therapist, among others at NeumoCesar institution.

The PR program at the Pneumological Center comprised three sessions a week for 3 months. Vital signs were monitored and recorded before, during, and after each session. The sessions were 50 min and included (1) 5 minutes of warm-up, (2) 20 min of aerobic exercise on a recumbent bike along with a respiratory stimulator comprising 3 balloons, 1200 ml. The respiratory stimulator was used for 4 series of 5 repetitions and adjusted to accommodate the tolerance of each patient; (3) 20 minutes of varied physical exercise including resistance, flexibility, and strength; (4) and 5 minutes of cool down.

**Post.**   Spirometry was performed after completing the rehabilitation program. The mMRC dyspnea questionnaire was also assessed.

## Respiratory function and dyspnea assessments

Respiratory function was assessed through spirometry tests performed using the MiniBox SN 2021102368-PulmOne system with easy one connects; Minibox version: 3.1.1.19 software was used to record patient data such as age, gender, height, and weight at the beginning and end of the rehabilitation program. Dyspnea was assessed with the mMRC scale (Modified Medical Research Council) defined by the Medical Research Council [16], which was used as a self-assessment tool to measure the degree of disability of the patient due to dyspnea in daily activities on a scale from 0 to 4 (Table 1).

## Statistical analysis

Differences in respiratory function by gender before and after therapy were evaluated using a univariate approach. The age ranges by gender of the patients who completed the PR program were compared using a Welch's t-test ($t$). The variability in FVC, FEV1, PEF, FEF 25–75, and FEV1/FVC ratio was compared using ANOVA (F = Fisher's F test; df = degrees of freedom; p = probability of significance). The variability in mMRC was analized using a non-parametric ANOVA, and a Kruskal-Wallis test ($H$) was used to measure the central tendency of the samples [17]. Whether the age, height, or weight of the patients affected the variability observed in

**Table 1. Parameters and criteria employed to evaluate respiratory function and dyspnea.**

| Respiratory function |
| --- |
| Forced vital capacity (FVC), expressed in liters, its normal values were ≥ 80% of the predicted value. |
| Forced expiratory volume in one second (FEV1), expressed in liters, and its normal value was ≥ 80% of the predicted value. |
| Ratio between forced expiratory volume in one second and forced vital capacity (FEV1/FVC) normal value > 70% of the predicted value. |
| Forced expiratory flow 25–75 (FEF25-75%), measured in liters, and its normal value > 65% of the predicted value [5]. |
| Peak expiratory flow (PEF), the maximal flow that can be exhaled, measured in liters. |
| **mMRC dyspnea scale** |
| 0 no dyspnea except during strenuous exercise |
| 1 shortness of breath when running on a level surface or up a slight incline |
| 2 walks slower than people of the same age on a level surface due to shortness of breath or has to stop to catch breath when walking at a normal pace on a level surface. |
| 3 stops to catch breath after walking ~100 m or after a few minutes on a level surface |
| 4 too out of breath to leave the house, or out of breath while dressing or undressing |

the respiratory function was explored through ANCOVA. Comorbidities by gender and the oxygen delivery system used were assessed for their affect the FVC variability observed before and after therapy using nested ANOVA. For all univariate analyses, assumptions of normality and homogeneity of variances were evaluated using a Shapiro–Wilk test and Levene test, respectively.

A multivariable approach was used to assess the correlation and discrimination capacity of the respiratory responses before and after therapy. A multidimensional scaling test (MDS) was conducted to determine which of the respiratory responses showed greater variability; the respiratory response that explains the differences observed indicates the success/failure of the intervention. MDS allows similarities (or distances) in respiratory function to be represented as points in a high dimensional space, which are reduced into a lower dimensional space, thereby providing a visual indication of the similarity (or dissimilarity) [18, 19]. It also permits the identification of whether one cluster of patients is dimensionally distinct from another via the identification of the respiratory function with the highest variability and discrimination degree without linearity assumptions or a priori clustering [20]. The goodness of fit between the fitted and observed distances was measured using "Kruskal's stress" (S) [20, 21], which is the average of the deviations between the end and the initial spatial distances normalized to take values between 0 and 1. Values near 1 indicate the worst fit, and values near 0 indicate the best fit. However, values between 0.025 and 0.05 are considered good values, < 0.025 are excellent, and values equal to 0 are perfect [19]. A linear discriminant analysis was conducted to identify respiratory responses with the greatest discrimination capacity between pre- and post-therapy performance.

Pearson's simple and multiple correlation tests were performed to assess whether severity and comorbidities could be associated with respiratory functions before and after therapy intervention. Pearson's simple correlation test was performed considering the presence/absence of comorbidities versus the severity of Covid-19 infection and the type of comorbidities versus the severity of Covid-19 infection. Pearson's multiple correlation test was carried out by testing the severity of Covid-19 disease versus respiratory function before and after therapy intervention.

Patients were classified into three groups based on the severity of symptoms according to the clinical course of Covid-19 infection [22]. Patients who did not receive oxygen therapy

were classified as having a mild illness. Patients who received low-flow oxygen therapy were categorized as having moderate illnesses. Those admitted to an intensive care unit (ICU) and received ventilatory support using non-invasive mechanical ventilation (NIMV) or invasive mechanical ventilation (IMV) systems were considered to have severe Covid-19 infection.

All statistical analyses were performed using the software Rwizard 4.3 [19] and the following R packages: car [21], hier.part [23], lawstat [24] nortest [25], overlap [26], stat [27], and usdm [28].

## Ethical considerations

Before being included in the study, patients who met the inclusion criteria had to give their written consent to participate in the study. Written informed consent was obtained from adult participants ($\geq$18 years). The risks and benefits associated with participation in the study were clearly explained to each patient. The study was conducted in accordance with the Declaration of Helsinki ethical guidelines for research in humans. Likewise, approval was obtained from the ethics committee of Manuela Beltrán University (TRCC-CE-I-2210009).

## Results

A total of 100 adult patients were included in the study (Men = 42; women = 58; ratio = 1:1.4). The age range was broad for both genders, and no significant difference in the age range was observed between them (t = 0.058, df = 80.36, P = 0.95; Table 2). Respiratory function differed

**Table 2. Descriptive data of the patients who completed the PR program and variables assessed.**

| Patients 'trait | Women (N = 42) | Men (N = 58) |
|---|---|---|
| **Body measurements** | | |
| Height (m) | 1.44–1.76 ($\bar{X}$ = 1.61) | 1.52–1.90 ($\bar{X}$ = 1.73) |
| Weight (kg) | 48–110 ($\bar{X}$ = 74) | 42–128 ($\bar{X}$ = 80) |
| Age | 22–86 ($\bar{X}$ = 53.6) | 19–81 ($\bar{X}$ = 52.8) |
| **Oxygen delivery system** | | |
| Invasive mechanical ventilation (IMV) | 6 | 10 |
| Non-invasive mechanical ventilation (NIMV) | 17 | 28 |
| Low flow | 9 | 8 |
| None | 10 | 12 |
| **Comorbidities** | | |
| Asthma | 6 | 3 |
| COPD | 5 | 2 |
| Diabetes | 1 | 2 |
| High blood pressure | 4 | 5 |
| Hypothyroidism | 2 | 0 |
| Obesity | 2 | 1 |
| OSA | 0 | 1 |
| Non-comorbidities | 22 | 45 |
| **Marital status** | | |
| None | 26 | 40 |
| Partner | 16 | 19 |
| **Health insurance** | | |
| Contributory | 38 | 51 |
| Contributory + Prepaid insurance | 1 | 2 |
| Subsidized | 3 | 6 |

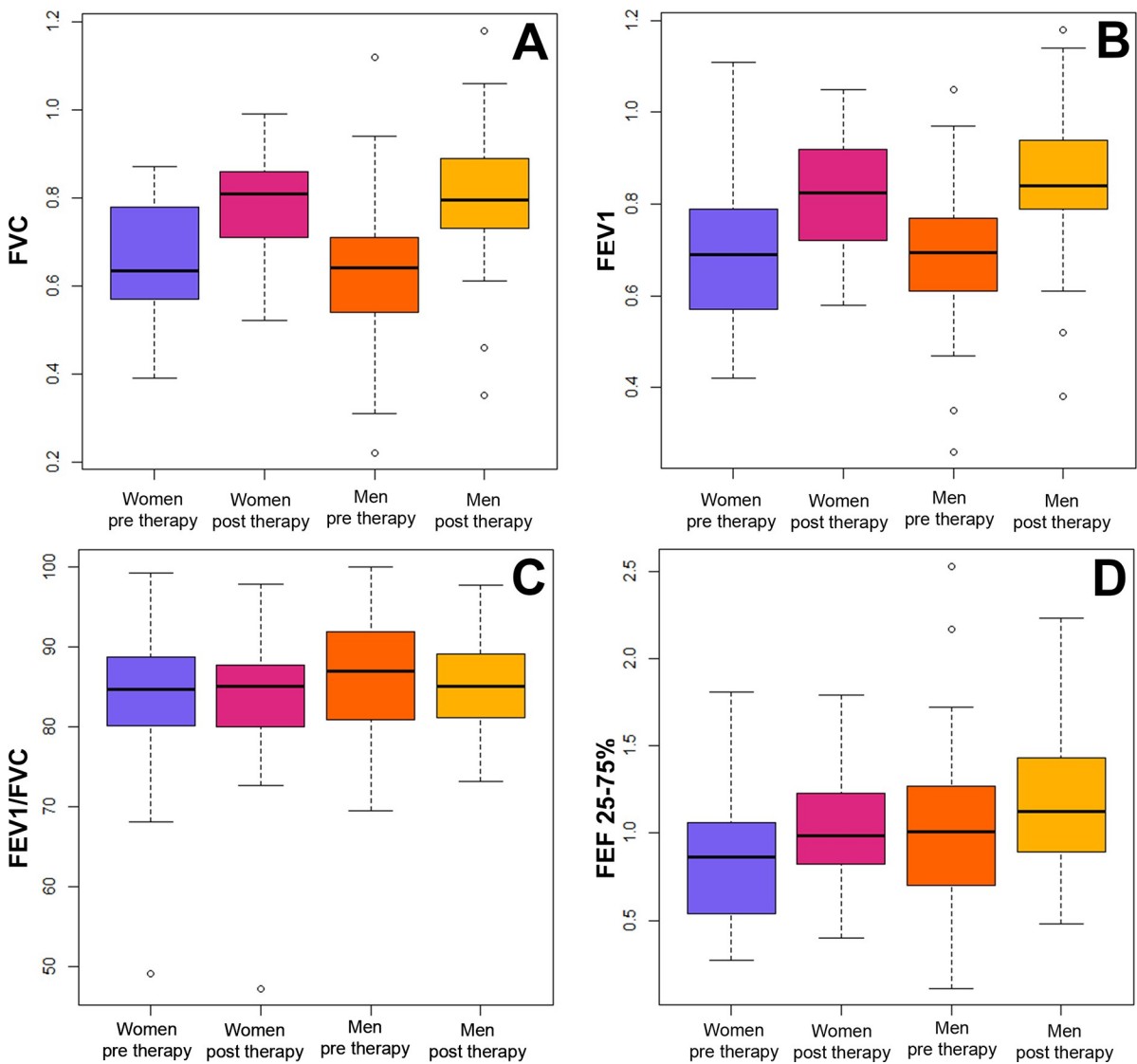

**Fig 1. Differences by gender in the respiratory function before and after therapy using a univariate approach.** (A-D) ANOVA for assessing respiratory responses before and after therapy. FVC = forced volume capacity; FEV 1 = forced expiratory volume; PEF = peak expiratory flow; FEF 25–75 = forced expiratory flow in 25 to 75% and FEV1/FVC1 ratio.

greatly before and after therapy; however, differences in pre- and post-therapy performance were low within each gender (Table 2). Men and women exhibited significant differences in FVC (F = 19.47, df = 3, P <0.0001), FEV1 (F = 22.49, df = 3, P = <0.0001), and mMRC (H = 99.022, df = 3, P < 0.0001) before and after therapy (Fig 1). However, no differences were observed in PEF and FEF 25–75%, as well as in the FEV1/FVC ratio (Fig 2). ANCOVA showed that age, height, or weight was not correlated with FVC, FEV1, or mMRC (Table 3), indicating that these variables had no effect on the significant differences observed before and after therapy.

Nested ANOVA indicated that the absence or presence of comorbidities in men and women showing significant differences in FVC was associated with the oxygen delivery system that had been used to treat SARS-CoV-2 infection (F = 22.49, df = 3, P = <0.0001); the IMV and NIMV systems were correlated with FVC values under 70%, and the low-flow system and

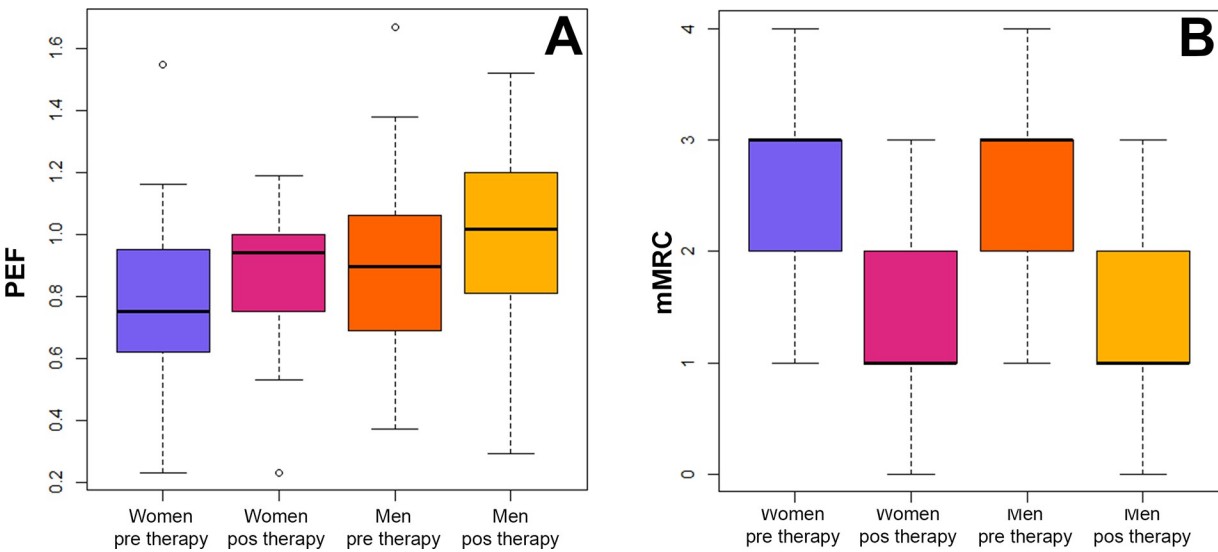

**Fig 2. Differences by gender in peak expiratory flow function and dyspnea perception before and after therapy.** (A) ANOVA for assessing the response of peak expiratory flow (PEF). (B) Non-parametric ANOVA using a Kruskal-Wallis test for assessing the perception of dyspnea. mMRC = Medical Research Council Dyspnea Scale.

non-oxygen delivery were correlated with FVC values over 70%. However, comorbidities did not affect differences observed between men and women in FVC and FEV1.

Pearson's simple correlation tests showed that there is no association between the severity of Covid-19 infection and the absence/presence of comorbidities (R = -0.058, P = 0.57) or type of comorbidities (R = 0.017, P = 0.87). However, Pearson's multiple correlation tests showed that the severity of Covid-19 infection was positively correlated with the FEV1/FVC ratio and FEF 25–75, and negatively with FEV1 before PR program. After PR program, the severity of Covid-19 infection was positively correlated with the FEV1/FVC ratio, FEF 25–75, and PEF, and negatively with mMRC (Table 4). Nevertheless, in all positive or negative associations, the proportion of explained variance was less than 50%, suggesting low to moderate correlations.

MDS revealed significant dissimilarities (S = 0.029) in the respiratory function of patients before and after therapy (Fig 3A), indicating that therapy had positive effects on the respiratory pathologies caused by SARS-CoV-2 infection. In addition, the MDS suggested that FVC, FEV1and mMRC are robust diagnostic indicators of the patient recovery process because these respiratory function variables were the most variable and had the highest discrimination degree. The linear discriminant analysis confirmed that the therapy had positive effects on patients recovering from respiratory pathologies caused by SARS-CoV-2 infection, as indicated by the pronounced differences observed before and after therapy for each gender (Fig 3B–3D). The first canonical axis explained 91.3% of the variance and was correlated with differences in FVC and FEV1; the second canonical axis explained 7.76% of the variance and was correlated with differences in mMRC and FEF 25–75 (Fig 3C). The linear discriminant

**Table 3. ANCOVA for assessing the effects of age, height, and weight of the patients on respiratory function variables that showed significant differences before and after therapy.**

| Respiratory output | Age | Height | Weight |
|---|---|---|---|
| FVC | F = 0.77, P = 0.37 | F = 0.95, P = 0.32 | F = 0.035, P = 0.85 |
| FEV1 | F = 0.007, P = 0.98 | F = 0.27, P = 0.60 | F = 0.031, P = 0.86 |

**Table 4. Correlation of severity of Covid-19 infection with respiratory functions before and after of PR program.** Upper diagonal part contains correlation coefficient estimates. Lower diagonal part contains corresponding p-values.

| | Severity | FEV1/FVC | FVC | FEV1 | PEF | FEF25-75 | mMRC |
|---|---|---|---|---|---|---|---|
| **Before PR program** | | | | | | | |
| Severity | ***** | 0.407 | -0.239 | -0.078 | 0.119 | 0.267 | -0.151 |
| FEV1/FVC | <**0.001** | ***** | -0.334 | 0.075 | 0.242 | 0.633 | -0.095 |
| FVC | **0.017** | **0.001** | ***** | 0.869 | 0.410 | 0.152 | -0.531 |
| FEV1 | 0.439 | 0.460 | <**0.001** | ***** | 0.537 | 0.492 | -0.638 |
| PEF | 0.240 | **0.015** | <**0.001** | <**0.001** | ***** | 0.600 | -0.405 |
| FEF25-75 | **0.007** | <**0.001** | 0.131 | <**0.001** | <**0.001** | ***** | -0.270 |
| mMRC | 0.133 | 0.346 | <**0.001** | <**0.001** | <**0.001** | **0.007** | ***** |
| **After PR program** | | | | | | | |
| Severity | ***** | 0.464 | -0.195 | -0.005 | 0.175 | 0.286 | -0.263 |
| FEV1/FVC | <**0.001** | ***** | -0.228 | 0.196 | 0.278 | 0.627 | -0.215 |
| FVC | **0.052** | 0.022 | ***** | 0.875 | 0.344 | 0.258 | -0.449 |
| FEV1 | 0.962 | 0.050 | <**0.001** | ***** | 0.510 | 0.611 | -0.578 |
| PEF | 0.081 | 0.005 | <**0.001** | <**0.001** | ***** | 0.503 | -0.572 |
| FEF25-75 | **0.004** | <**0.001** | **0.009** | <**0.001** | <**0.001** | ***** | -0.436 |
| mMRC | **0.008** | **0.032** | <**0.001** | <**0.001** | <**0.001** | <**0.001** | ***** |

FVC = forced volume capacity; FEV 1 = forced expiratory volume; PEF = peak expiratory flow; FEF 25–75 = forced expiratory flow in 25 to 75% and FEV1/FVC1 ratio. mMRC = Medical Research Council Dyspnea Scale.

analysis also yielded the same diagnostic indicators for the patient recovery process as were identified in the ANOVA and MDS. However, the cross-validation percentage was moderate (55.5%), indicating that large patient sampling is required for the discrimination function to achieve a higher classification accuracy of the respiratory function before and after therapy.

## Discussion

The results of this study provide evidence that PR program has positive effects on both men and women patients recovering from respiratory pathologies caused by SARS-CoV-2 infection. Significant differences in lung function parameters such as FVC, FEV1, and mMRC were observed before and after therapy. Age, height, nor weight was correlated with significant differences in FVC, FEV1, and mMRC, indicating that the therapy provided benefits to all patients regardless of their physical characteristics.

FEV1 can be used to evaluate the degree of airway obstruction when used in conjunction with FEV1/FVC, which confirms the presence of obstruction. No significant differences were observed in FEV1/FVC before and after therapy, which confirms the presence of a restrictive pulmonary pattern and the absence of an obstructive pulmonary pattern, and changes in FEV1 before and after therapy were significant. Restrictive patterns are characteristic of pulmonary disorders that result in damage to the lung epithelium [15]. The significant change in FVC reflects an improvement in pulmonary function following therapy, and this change was associated with the restrictive pattern caused by SARS-CoV-2 infection. However, it is recommended for future studies to complement the evaluation of FVC with measurement of lung volumes.

mMRC measures the perception of dyspnea, and it significantly improved after the therapy. This probably stemmed from the strengthening of the respiratory muscles. Rehabilitation also improves lung volume, oxygenation, and thus the perception of dyspnea. Resistance training

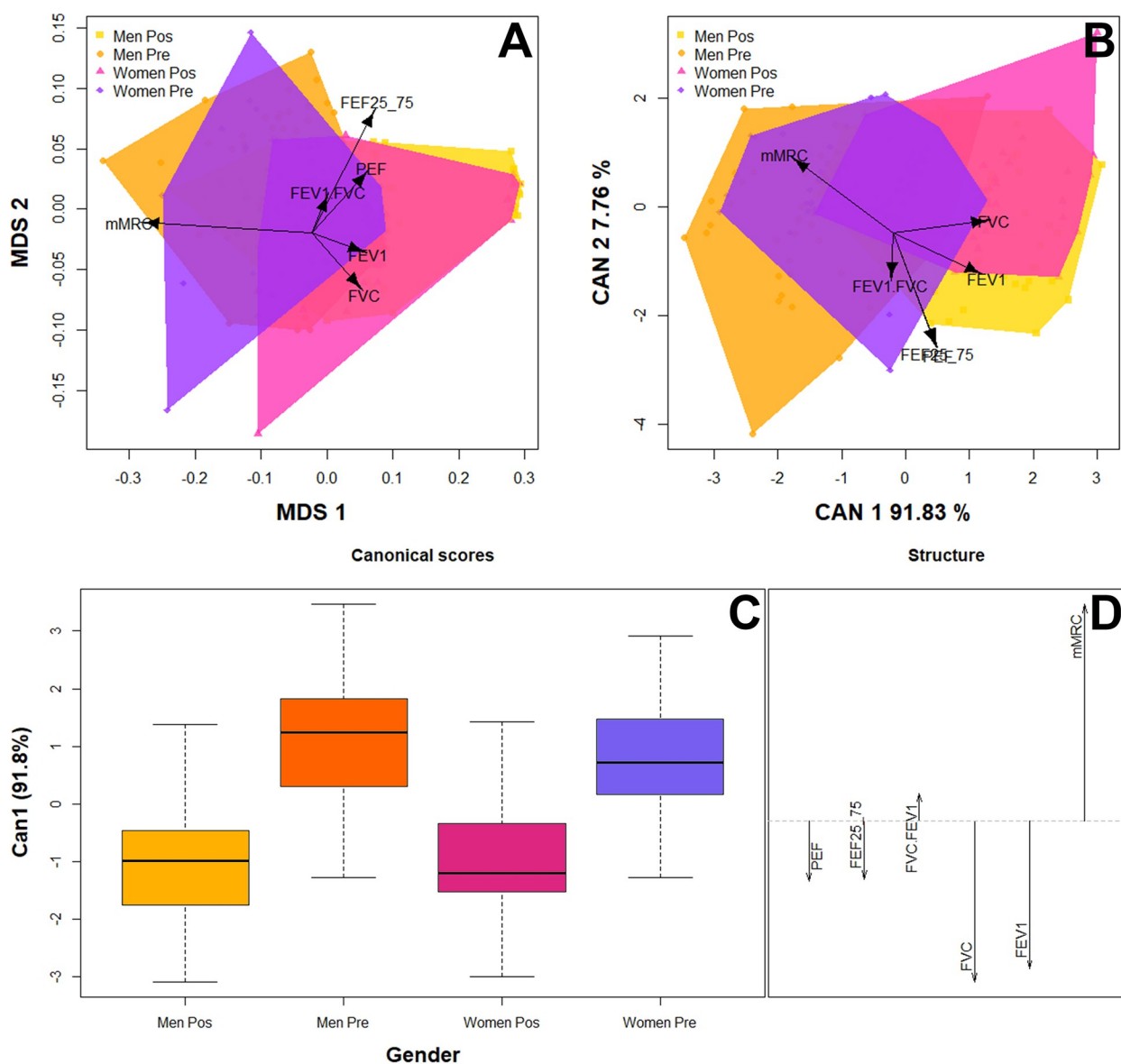

**Fig 3.** (A) Multidimensional scaling test (MDS) assessing the respiratory function of patients before and after therapy. The goodness of fit is indicated by the "Kruskal's stress (S)." Values near 1 indicate the worst fit, and values near 0 indicate the best fit. Values between 0.025 and 0.05 are considered good values, values < 0.025 are excellent, and values equal to 0 are perfect. (B-D) Linear discriminant analysis assessing the respiratory responses of patients before and after therapy. (B) Biplot depicting the observed variability in respiratory function of patients before and after therapy. (C) Canonical scores and structure of the linear discriminant analysis. (D) The length of the vector denotes the discrimination capacity of each respiratory function. FVC = forced volume capacity; FEV 1 = forced expiratory volume; PEF = peak expiratory flow; FEF 25–75 = forced expiratory flow in 25 to 75% and FEV1/FVC1 ratio. mMRC = Medical Research Council Dyspnea Scale.

during the early phases of rehabilitation likely improved the mechanical efficiency of the lungs; the observed improvement in lung function variables was similar to changes observed in patients undergoing rehabilitation programs following a diagnosis of COPD [29].

Our results are consistent with previous studies showing significant improvements in FVC, FEV1, and mMRC in Covid-19 patients after PR [30–33]. Our findings are also consistent with the results of Liu et al. [12]. In this study, a randomized controlled trial was conducted with 72 participants in Huanggang, China, and changes in lung function were evaluated.

Significant differences in some of the variables measured in our study, such as FVC, FEV1, and FEV1/FVC, before and after rehabilitation were detected. They also identified significant changes in DLCO before and after rehabilitation; however, these variables were not measured in our study due to the limited technological resources available at the research institution. This is a common limitation in research conducted in developing regions, where access to advanced diagnostic tools may be restricted.

The use of both invasive and non-invasive positive pressure ventilatory support during hospitalization was significantly associated with FVC lower than 70% compared with the group that did not use this support, which had FVC values higher than 70% (i.e., patients categorized with severe infection in the severity criteria). This might stem from the severity of Covid-19, which is directly related to the need to use this type of support [34]. These results indicate that positive pressure ventilatory support might be associated with the deleterious effects on lung tissue. These could be secondary to the difficulty of titrating the ventilatory parameters due to the pulmonary heterogeneity caused by the disease [35]. Furthermore, our findings suggest that the severity of Covid-19 infection may also play a role in these deleterious effects, as positive pressure ventilatory support was significantly associated with lower forced vital capacity (FVC) in patients with severe infection compared to those with milder disease. In both cases, the lung tissue damage might be related to self-induced lung injury (P-SILI) or ventilator-induced lung injury (VILI) [35]. These findings suggest that positive pressure ventilatory support, both invasive and non-invasive, may have had deleterious effects on pulmonary function, potentially contributing to prolonged patient recovery. However, it is important to acknowledge that the observed impact on pulmonary function could be attributed to both the severity of Covid-19 infection and the use of positive pressure ventilatory support. The presence of severe disease in these patients could have contributed to impaired lung function regardless of ventilatory support. Therefore, the observed difference in recovery rates among patients who received positive pressure ventilatory support and those who did not may be due to a combination of factors, including the severity of the disease and the effects of the support itself. The FVC, FEV1and mMRC parameters were robust diagnostic indicators of the patient recovery process, given that these respiratory responses were the most variable and had the highest discrimination capacity.

While a significant change was observed in FEV1, a variable commonly used to assess the severity of obstructive patterns, no significant change was detected in the FVC/FEV1 ratio. As a result, we cannot conclude that there are clinically relevant changes related to obstructive patterns. Nevertheless, further studies are warranted to investigate the specific changes in lung parenchyma induced by Covid-19 infection both before and after rehabilitation evaluating small and large airway function. These studies could potentially shed light on the long-term pulmonary alterations that may arise from Covid-19 infection [36–39]. The long-term impacts of the sequelae of Covid-19 also merit further investigation; the use of a large sample of patients is needed to improve the ability to detect changes in respiratory function before and after therapy.

It was found that PR had positive effects on recovery from respiratory pathologies caused by SARS-CoV 2 infection and that it accelerated the recovery process. However, which techniques are optimal and how long the rehabilitation program should last remain unclear given that benefits have been observed in 2-week programs [30, 40]. Other studies have reported benefits in 6-week programs [29]; in our study, positive effects were observed in a 12-week program [41].

PR accelerated the recovery of lung function and the perception of dyspnea in patients diagnosed with Covid-19. However, additional studies are needed to clarify whether rehabilitation method and program duration affect the benefits observed; the effectiveness of remote

rehabilitation procedures such as Telemedicine also merits further study. Given the difficulty of access to medical facilities in Colombia, special effort is needed to ensure equal access to comprehensive care.

The generalizability of the findings may be limited by the relatively small sample size and the single-center design. The sample, drawn from a single institution, may not be fully representative of the broader population, potentially restricting the applicability of the results to other settings. While the data provides valuable insights, future prospective studies with larger, more diverse samples are needed to confirm these findings and explore the long-term effects of the intervention, controlling for potential confounding variables.

## Conclusions

- A 12-week PR program resulted in significant changes in lung function in Colombian patients.

- The most pronounced changes were observed in the perception of dyspnea, FVC, and FEV1. Age, height, weight, and the presence of comorbidities were not correlated with significant changes observed between groups.

- Patients with a history of using positive pressure devices such as NIMV and IMV during hospitalization exhibited FVC values below 70%, a pattern associated with restrictive lung diseases. Further research is warranted to determine whether these changes are attributable to the use of positive pressure ventilation, the severity of the disease, or a combination of both.

- FVC and mMRC are robust diagnostic indicators of the patient recovery process given that these respiratory responses were the most variable and had the highest discrimination capacity.

## Supporting information

**S1 File.**
(XLSX)

**S2 File.**
(XLSX)

## Acknowledgments

We thank the Neumocesar Pneumological Center and Doctor Edinson Valencia for their constant support and commitment to this research. We thank Christopher K Akcali for revising a draft of the manuscript.

## Author Contributions

**Conceptualization:** Carlos D. Páez-Mora, Diana Carolina Zona, Teddy Angarita-Sierra, Matilde E. Rojas-Paredes, Daniela Cano-Trejos.

**Data curation:** Carlos D. Páez-Mora, Diana Carolina Zona, Teddy Angarita-Sierra, Matilde E. Rojas-Paredes, Daniela Cano-Trejos.

**Formal analysis:** Diana Carolina Zona, Teddy Angarita-Sierra.

**Investigation:** Carlos D. Páez-Mora, Teddy Angarita-Sierra.

**Methodology:** Diana Carolina Zona, Teddy Angarita-Sierra.

**Project administration:** Diana Carolina Zona.

**Writing – original draft:** Carlos D. Páez-Mora, Diana Carolina Zona.

**Writing – review & editing:** Diana Carolina Zona, Teddy Angarita-Sierra.

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
