## [Decision Letter · Decision Letter 0]

10 Jun 2024

PONE-D-24-08060Changes in lung function and dyspnea perception in Colombian Covid-19 patients after a 12-week pulmonary rehabilitation program.PLOS ONE

Dear Dr. Zona,

Thank you for submitting your manuscript to PLOS ONE. After careful consideration, we feel that it has merit but does not fully meet PLOS ONE’s publication criteria as it currently stands. Therefore, we invite you to submit a revised version of the manuscript that addresses the points raised during the review process. Your manuscript has been evaluated by two reviewers, and their comments are appended below.

The reviewers have commented on the study design, conclusions, and the statistical analysis, among their other comments. Please ensure you address each of the reviewers' comments when revising your manuscript.

We look forward to receiving your revised manuscript.

Kind regards,

Hugh Cowley

Staff Editor

PLOS ONE

Reviewers' comments:

Reviewer's Responses to Questions

**Comments to the Author**

1. Is the manuscript technically sound, and do the data support the conclusions?

Reviewer #1: Partly

Reviewer #2: Partly

2. Has the statistical analysis been performed appropriately and rigorously? 

Reviewer #1: Yes

Reviewer #2: Yes

3. Have the authors made all data underlying the findings in their manuscript fully available?

Reviewer #1: No

Reviewer #2: No

4. Is the manuscript presented in an intelligible fashion and written in standard English?

Reviewer #1: Yes

Reviewer #2: Yes

5. Review Comments to the Author

Reviewer #1: This study involved rehabilitation programs that had positive effects on respiratory function after SARS-CoV-2 infection ,Here are my comments

1.The inclusion criteria did not discriminate between degree of COVID-19 infection which could affect the degree of lung injury.If the patients had only mild diseases,the recovery can happen spontaneously without rehabilitation

2. The FEF 25-75% was nit different between both groups.Why the author conclude that this might be associated with changes in the small airway

3.Tle study did not measure previous lung function before COVID-19 infection,therefore how can you conclude that the cases that didnot improve could be explained by underlying lung disorders.

4,It will be better to perform DLCO test in addition

Reviewer #2: Your study is sound and practical about ventilation effectiveness and complications, and demistify exclusion criteria based on comorbidities and demographics. Intensivists and pathologists in other countries also reported alveoli damage depending on the parameters. I noticed the issues below that if addressed will improve the quality of your paper:

Sample size n=100 with N=231 corresponds to e=7.5% instead of 5%, either adjust CI or run the analysis with n=146;

Normalize VMN/VMNI, IV/NIV and IMV/NIMV abbreviations, also variables e.g. mMRC/MRC;

Missing decimal point in Height's P in Table 3;

Two plots named FEV1 in Figure 1;

Figure 3 is difficult to read with color blindness

See https://stackoverflow.com/questions/57153428/r-plot-color-combinations-that-are-colorblind-accessible ;

Add summary statistics for spirometry variables pre and post intervention (e.g. in Table 2), supporting the lack of obstructive patterns before PR (in Discussion);

Literature has retrospective studies of Covid-19 patients 10+ months after discharge, with radiology evaluations and other PR methods. Did you thought of extending your study or adding another cohort (e.g. without the 100% PR compliance inclusion criteria) ?

6. PLOS authors have the option to publish the peer review history of their article (what does this mean?). If published, this will include your full peer review and any attached files.

Reviewer #1: No

Reviewer #2: No

---

## [Author Response · Author response to Decision Letter 0]

18 Jul 2024

Dear Editor,

We wish to respond to the reviewers of our research article entitled “Changes in Lung Function and 

Dyspnea Perception in Colombian COVID-19 Patients After a 12-Week Pulmonary Rehabilitation Program.” 

Below, we address each point. It is important to clarify that these changes have been applied to the 

manuscript.

1. Is the manuscript technically sound, and do the data support the conclusions?

• Coherence errors between sections have been corrected.

2. Have the authors made all data underlying the findings in their manuscript fully available?

• All files containing the raw data have been uploaded as supplemental material in Excel 

format.

3. This study involved rehabilitation programs that had positive effects on respiratory function 

after SARS-CoV-2 infection. Here are my comments: The inclusion criteria did not discriminate 

between the degree of COVID-19 infection, which could affect the degree of lung injury. If the 

patients had only mild disease, recovery could occur spontaneously without rehabilitation.

• Severity criteria for SARS-CoV-2 infection have been incorporated into the methodology to 

support the findings of the study pre- and post-pulmonary rehabilitation.

4. The FEF 25-75% was not different between both groups. Why do the authors conclude that this 

might be associated with changes in the small airways?

• This point has been corrected.

5. The study did not measure previous lung function before COVID-19 infection; therefore, how can 

you conclude that the cases that did not improve could be explained by underlying lung 

disorders?

• Comorbidities were included as variables in the analysis. ANCOVA analysis showed no 

correlation between age, height, weight, and lung function measures (FVC, FEV1, MRC). 

Nested ANOVA indicated that the presence or absence of comorbidities did not affect the 

differences observed in FVC and FEV1 between men and women. The text has been 

adjusted accordingly. Indeed, we do not have previous data on lung function before 

COVID-19 infection.

6. It would be better to perform DLCO testing in addition.

• DLCO testing was not performed because the research institution lacked this diagnostic 

capability, due to the technological availability in the Colombian region where the research 

was conducted.

7. Your study is sound and practical regarding ventilation effectiveness and complications, and it 

demystifies exclusion criteria based on comorbidities and demographics. Intensivists and 

pathologists in other countries also reported alveolar damage depending on the parameters. I 

noticed the issues below that, if addressed, will improve the quality of your paper: Sample size 

n=100 with N=231 corresponds to e=7.5% instead of 5%; either adjust CI or run the analysis with 

n=146.

• This point has been corrected.

8. Normalize VMN/VMNI, IV/NIV, and IMV/NIMV abbreviations, as well as variables, e.g., 

mMRC/MRC.

• This point has been corrected.

9. Missing decimal point in Height's p-value in Table 3.

• This point has been corrected.

10. Two plots are named FEV1 in Figure 1.

• This point has been corrected.

11. Figure 3 is difficult to read for individuals with color blindness.

• All figures have been reedited. We employed the IBM Color Blind Safe Palette as follows: 

Ultramarine 40, Indigo 50 (#785EF0), Magenta 50 (#DC267F), Orange 40 (#FE6100), Gold 

20 (#FFB000).

12. Add summary statistics for spirometry variables pre- and post-intervention (e.g., in Table 2).

• We uploaded a file with the summary statistics for spirometry variables pre- and post-intervention as supplemental material.

13. Supporting the lack of obstructive patterns before PR (in Discussion).

• While we acknowledge the importance of expanding the study as suggested, it is not 

possible to do so due to financial limitations. This is a common challenge in Latin America. 

However, we consider the study findings to be relevant as they demonstrate the effects of 

pulmonary function before and after a rehabilitation program in a Latin American region.

Finally, we would like to request that the order of the authors presented in the manuscript be maintained. 

At the time of submitting the manuscript, the page did not allow selecting who is the corresponding author 

and who is the first author.

---

## [Decision Letter · Decision Letter 1]

24 Sep 2024

PONE-D-24-08060R1Changes in lung function and dyspnea perception in Colombian Covid-19 patients after a 12-week pulmonary rehabilitation program.PLOS ONE

Dear Dr. Zona,

Thank you for submitting your manuscript to PLOS ONE. After careful consideration, we feel that it has merit but does not fully meet PLOS ONE’s publication criteria as it currently stands. Therefore, we invite you to submit a revised version of the manuscript that addresses the points raised during the review process.

We look forward to receiving your revised manuscript.

Kind regards,

Mehrnaz Kajbafvala, Ph.D

Academic Editor

PLOS ONE

Journal Requirements:

Reviewers' comments:

Reviewer's Responses to Questions

**Comments to the Author**

1. If the authors have adequately addressed your comments raised in a previous round of review and you feel that this manuscript is now acceptable for publication, you may indicate that here to bypass the “Comments to the Author” section, enter your conflict of interest statement in the “Confidential to Editor” section, and submit your "Accept" recommendation.

Reviewer #1: All comments have been addressed

Reviewer #3: (No Response)

2. Is the manuscript technically sound, and do the data support the conclusions?

Reviewer #1: Yes

Reviewer #3: Partly

3. Has the statistical analysis been performed appropriately and rigorously? 

Reviewer #1: Yes

Reviewer #3: Yes

4. Have the authors made all data underlying the findings in their manuscript fully available?

Reviewer #1: Yes

Reviewer #3: Yes

5. Is the manuscript presented in an intelligible fashion and written in standard English?

Reviewer #1: Yes

Reviewer #3: No

6. Review Comments to the Author

Reviewer #1: the authors have corrected the manuscript as I suggested and should be accpted for publication.however,the corrected part should be highlight to be easier to read

Reviewer #3: Please address these questions:

Abstract:

>For the first time, please use the two abbreviations PR and mMRC in full form.

>In the result section you should mention the exact p-value for statistical significance.

Line 35: “A large sample of patients is needed to clarify the effects of therapy on respiratory function” is the limitation of the study and there is no need to mention it in the conclusion of the abstract.

Introduction:

Lines 80-81: “The Ethical Guidelines for Human Research from the Declaration of Helsinki were followed, and written informed consent was obtained for participation.”, You should put this sentence in the method section instead of the introduction section.

>Introduction is too long, please omit some unnecessary sentences.

Materials and methods:

>Please use FEF and PEF in full for the first time and then use them as abbreviations throughout the manuscript.

Line 118: ”60 to 80% of the maximum cardiac capacity.”, Please also mention in this section the intensity of the exercises used; mild, moderate, sever.

Lines 126-130: Training sessions did not include warm-up? A training session should include both warm-up and cool-down.

> The word “we” has been used a lot through the manuscript, please remove some of them and write the sentence in a passive form.

Lines 181-186: If there is any reference for classifying the severity of the symptoms, please add.

Discussion:

Please clarify the limitations of your study at the end of this section.

7. PLOS authors have the option to publish the peer review history of their article (what does this mean?). If published, this will include your full peer review and any attached files.

Reviewer #1: No

Reviewer #3: No

---

## [Author Response · Author response to Decision Letter 1]

7 Oct 2024

Dear Editor’s,

Thank you for the opportunity to revise our manuscript titled “Changes in Lung Function and 

Dyspnea Perception in Colombian COVID-19 Patients After a 12-Week Pulmonary 

Rehabilitation Program.” We appreciate the valuable feedback from you and the reviewers, 

which has helped us enhance the quality of our work. Below, we address each point raised by 

the reviewers.

Abstract

1. Use of full forms for PR and mMRC: Thank you for your suggestion. We have revised 

the manuscript to include the full forms of PR (pulmonary rehabilitation) and mMRC 

(modified Medical Research Council) the first time they appear.

2. Mentioning exact p-values: We acknowledge that Multidimensional Scaling (MDS) 

and Linear Discriminant Analysis (LDA) do not provide p-values, as they are 

classification techniques rather than hypothesis-testing methods. However, we have 

included exact p-values for all relevant univariate analyses in the results section to 

enhance clarity.

3. Limitation about sample size in the abstract conclusion: We have removed the 

statement regarding the need for a larger sample size from the conclusion of the 

abstract, recognizing it as a limitation more suitable for the discussion section.

Introduction

1. Ethical guidelines statement: We have moved the sentence regarding the Ethical 

Guidelines for Human Research from the Declaration of Helsinki from the introduction 

to the methods section, as suggested.

2. Omitting unnecessary sentences: We have shortened the introduction by removing 

the following sentences:

o “Which involves evaluating the mechanical properties of the respiratory 

system through volumes and capacities; it provides relevant information for 

the classification of both obstructive and restrictive disorders.”

o “Which has motivated evaluations of lung function to characterize changes in 

the lung parenchyma and assess the need for treatment.”

Materials and Methods

1. Use of FEF and PEF in full: We have made the requested adjustments and now use 

FEF (forced expiratory flow) and PEF (peak expiratory flow) in full the first time they 

appear.

2. Intensity of exercises: The intensity of exercises has been specified in the revised 

manuscript.

3. Warm-up and cool-down periods: We have rectified the oversight regarding warm-up 

and cool-down periods, ensuring that training sessions now include both.

4. Reducing use of "we": We have thoroughly revised the manuscript to reduce the 

frequency of the word "we" and adopted a more passive voice, making the writing 

more objective.

5. Reference for symptom severity classification: We have added the appropriate 

reference to support the classification of symptom severity.

Discussion

1. Clarifying limitations: We have clarified the limitations of our study at the end of the 

discussion section, as suggested.

Attached, we are sending the unmarked version of the document, as well as the version with 

track changes to facilitate the review and reading of the changes. We believe these revisions 

address all the concerns raised by the reviewers. 

Thank you for your consideration of our manuscript. We look forward to your positive 

response.

---

## [Editor Report · Decision Letter 2]

23 Oct 2024

Changes in lung function and dyspnea perception in Colombian Covid-19 patients after a 12-week pulmonary rehabilitation program.

PONE-D-24-08060R2

Dear Dr. Diana Carolina Zona

We’re pleased to inform you that your manuscript has been judged scientifically suitable for publication and will be formally accepted for publication once it meets all outstanding technical requirements.

Kind regards,

Mehrnaz Kajbafvala, Ph.D

Academic Editor

PLOS ONE
---

## [Editor Report · Acceptance letter]

5 Nov 2024

PONE-D-24-08060R2 

PLOS ONE

Dear Dr. Zona, 

I'm pleased to inform you that your manuscript has been deemed suitable for publication in PLOS ONE. Congratulations! Your manuscript is now being handed over to our production team.

Kind regards, 

on behalf of

Dr. Mehrnaz Kajbafvala 

Academic Editor

PLOS ONE